# PI3Kdelta-driven expansion of regulatory B cells impairs protective immune responses to *Trypanosoma congolense* parasite infection

Folayemi Olayinka-Adefemi[1], Xun Wu[1], Sen Hou[1], Milad Sabzevary-ghahfarokhi[1], Michelle N. Wray-Dutra[2,3], David J. Rawlings[2,3,4], Jude Uzonna[1], Aaron J. Marshall[1]*

1 Department of Immunology, Max Rady College of Medicine, Rady Faculty of Health Sciences, University of Manitoba, Winnipeg, Manitoba, Canada, 2 Seattle Children's Research Institute, Seattle, Washington, United States of America, 3 Departments of Pediatrics and Immunology, University of Washington School of Medicine, Seattle, Washington, United States of America, 4 Center for Immunity and Immunotherapies, Seattle Children's Research Institute, Seattle, Washington, United States of America

* aaron.marshall@umanitoba.ca

## Abstract

Phosphatidylinositol 3-kinase delta (PI3KCD) is a critical signaling enzyme for B cell development, activation, function and immune regulation. Gain-of-function mutations in PI3KCD result in the congenital immunodeficiency known as Activated PI3KCD Syndrome (APDS). APDS patients are prone to repeated infections and other serious clinical manifestations. Here, we determine how B cell-intrinsic expression of the APDS-associated PI3KCD$^{E1021K}$ mutation impacts immune responses to the protozoan parasite *Trypanosoma congolense*. PI3KCD$^{E1021K/B}$ mice exhibit a significant expansion of IL10-expressing B cells within the spleen and peritoneal cavity, which was associated with impaired control of *T. congolense* infection. Despite the generation of robust germinal center, plasma cell and antibody responses, PI3KCD$^{E1021K/B}$ mice show elevation in the first wave of parasitemia and increased mortality. We further characterize the phenotype of the expanded IL10-producing B cell population in PI3KCD$^{E1021K/B}$ mice, which show hallmarks of innate-like regulatory B cells (Breg) and expression of multiple inhibitory molecules. This Breg expansion is associated with reduced IFNγ/IL10 ratio, reduced TNFα production and impaired activation of myeloid cells, likely compromising the innate response to infection. These findings highlight the profound impact of dysregulated PI3KCD activity on regulatory B cells that can functionally impair innate immune responses controlling a systemic parasite protozoan disease.

## Author summary

B cells and antibodies play a critical role in the immune response to Trypanosome parasites. Molecular signaling networks within B cells can control the type of response generated during infection. Here, we studied how a genetic

**Data availability statement:** All relevant data are within the manuscript and its Supporting Information files.

**Funding:** AM was supported by an operating grant from the Canadian Institute of Health Research (CIHR), grant number: PJT-162268. https://www.cihr-irsc.gc.ca/e/193.html FOA was supported by Research Manitoba studentship and a University of Manitoba Graduate Student Fellowship. https://umanitoba.ca/gradu-ate-studies/sites/graduate-studies/files/2024-02/2024-25_UMGF-Award-Holders-Guide.pdf DR was supported in part by the Children's Guild Association Endowed Chair in Pediatric Immunology (to D.J.R), the Hansen Investigator in Pediatric Innovation Endowment (to D.J.R.) The funders had no role in study design, data collection and analysis, decision to publish, or preparation of the manuscript.

**Competing interests:** The authors have declared that no competing interests exist.

variant in the signaling enzyme PI3KCD, previously linked to human immune deficiencies, impacts B cell responses to Trypanosome infection. We find that mice expressing the PI3KCD$^{E1021K}$ mutation in their B cells show impaired control of Trypanosome infection, and alterations in several aspects of the immune response. Specifically, we noted these mice poorly control parasite growth within the first week of infection, a timeframe where specific antibody responses have not yet been generated. We noted an altered balance between pro-inflammatory and anti-inflammatory cytokine mediators produced within the first week of infection. This was associated with high numbers of regulatory B cells expressing multiple molecules capable of inhibiting other cells of the immune system. We further found that these mice show functional alterations in other critical immune cell types, such as macrophages and T cells. These findings highlight the impact of dysregulated PI3KCD activity on regulatory B cells that can impair immune responses controlling a systemic parasite protozoan disease.

## Introduction

Patients with activating germline mutations in the PI3KCD (p110δ) gene [1–3] have been recognized to present with complex immune characteristics such as altered B and T cell development, dysregulated antibody responses and a predisposition to recurrent respiratory infections [4,5]. Mice expressing the most common mutation (PI3KCD$^{E1021K}$) only in B cells demonstrated aberrant B cell development and activation, including elevated IL10 expression and expansion of innate-like B cell subsets [6]. Regulatory B cells (Breg) are a functionally defined population based on their capacity to secrete the anti-inflammatory molecule interleukin 10 (IL10) [7,8]. B cells can also utilize other suppressive molecules like TGF-β, IL35 [9–11] and surface proteins such as PD-L1 and CD1d [12,13]. Generation of IL10-producing B cells requires BCR signaling, CD40 as well as TLR [14–17] and can involve multiple B cell subsets including innate-like B cells and plasma cells [18,19]. Currently, there is no specific marker to identify Breg cells in humans or mice hence, they are identified by phenotypically shared characteristics with other B cells in addition to their capacity for IL10 production.

Bregs are important in health and disease and they play a significant role in immune tolerance and regulation of autoimmunity [20,21]. They can suppress autoimmunity in animal models such as collagen-induced arthritis and spontaneous colitis, and can modulate cancer and allergic asthma [22–24]. In murine models, B cell-derived IL10 was essential for reducing inflammation and autoimmunity [25]. In the context of infection, murine B cell-derived IL10 is shown to be effective in immune counter-regulation during cytomegalovirus infection by restricting excessive plasma cell expansion and the amplitude of virus-specific CD8$^+$ T cells in response to the disease [26]. While Bregs curb immunopathology via IL10 and other contact-dependent mechanisms, this same restraint can blunt optimal microbicidal responses. In malaria, elevated IL10 (including from B cells) limits Th1

effector activity and is linked to slower development of protective humoral immunity, illustrating a disease-modifying but double-edged role [27]. In Schistosoma helminth infection, Bregs arise in humans and mice, produce IL10, and suppress excessive inflammation and allergy, demonstrating potent immunoregulation that can benefit the host yet potentially permit parasite persistence [28]. In respiratory syncytial virus (RSV) infection, the presence of CD5+IL10-producing human cord blood B cells inhibited host anti-viral responses, leading to poor clinical outcomes [29]. Hyper-activation of PI3KCD selectively in B cells mediated the expansion of IL10+CD19+B220lo Bregs that impaired host immune responses to *Streptococcus pneumoniae* infection in the lungs [6]. The phenotype and functions of these PI3KCD-driven IL10-producing B cells are still being characterized, and the functional impact on immune responses to other types of pathogens remains to be investigated.

The protozoan parasite *Trypanosoma congolense* is an important animal pathogen, while other species such as *T. brucei* and *T. cruzi* cause African and American trypanosomiasis (Chagas disease) respectively in humans [30–32]. This protozoan parasite causes a disseminated systemic disease requiring prompt macrophage response and a robust activation of antigen-specific humoral immune responses for clearance [33]. Prior studies have indicated that PI3KCD has a role in promoting both immune regulatory and protective responses that impact the control of protozoan parasites, such as *Leishmania major* and *Trypanosoma congolense* [34,35]. We previously found that while germline PI3KCD loss-of-function mutant mice (p110δD910A) were unable to control *Trypanosoma congolense* infection due to the inability to mount effective antibody responses, the early control of parasitemia was paradoxically improved in these mice [34]. The latter finding was associated with a substantial reduction in IL10-producing innate-like B cells and an elevated IFNγ/IL10 ratio [34]. While B cell-intrinsic PI3KCD hyperactivation was shown to reciprocally drive expansion of IL10-producing innate-like B cells in spleen and lungs [4,6], it is unclear how this may impact the immune response to systemic protozoan parasites.

In this study, we have explored mechanisms by which B cell–intrinsic PI3KCD hyperactivation drives susceptibility to *Trypanosoma congolense* infection. We found that the B cell-intrinsic PI3KCD activation exacerbates the severity in early infection. We show that this susceptibility is associated with a significantly expanded population of innate-like Breg cells, altered balance in pro- and anti-inflammatory mediators and impaired myeloid cell function, which is critical for disease clearance. This study provides new insights into the unique roles of regulatory B cells in immune response and how dysregulation in their function may contribute to the pathogenesis of systemic protozoan diseases.

## Results

### PI3Kδ hyperactivation in B cells increases susceptibility to *T. congolense* infection

Prior studies have indicated that the signal transduction enzyme PI3Kδ (PI3KCD) has a role in promoting both immune regulatory and protective responses that impact the control of protozoan parasites, such as *Leishmania major* and *Trypanosoma congolense* [34,35]. Both loss-of-function and gain-of-function mutations in the catalytic subunit of PI3Kδ (PI3KCD) can impact B cell functions [36,37], indicating that appropriately regulated levels of PI3KCD activity are critical for normal humoral immunity. Here, we utilized a model of B cell-specific PI3KCD hyper-activation to explore how dysregulation of this pathway in B cells impacts the immune response to *T. congolense* infection. Mice bearing a conditional PI3KCDE1021K allele [4] were crossed with Mb1-Cre to introduce into B cells a gain-of-function (GOF) mutation functionally equivalent to human mutations causing activated PI3Kdelta syndrome [38,39]. These PI3KCDE1021K x MB1-Cre mice (abbreviated PI3KCDGOF/B), or Cre-negative littermate controls, were infected with 1000 *T. congolense* parasites intraperitoneally and monitored for parasitemia and survival. The results show that the PI3KCDGOF/B mutants developed a significantly higher parasite burden during the peak of infection between days 7–11 (Fig 1A) and exhibited approximately 30% mortality during the first two weeks post-infection (Fig 1B). PI3KCDGOF/B mice that survived the initial wave of parasitemia appear to control parasitemia similarly to controls, and no further mortality was observed. Serum biochemistry analyses revealed that PI3KCDGOF/B mice had elevated serum creatinine and alkaline phosphatase (ALP) at day 7 post-infection,

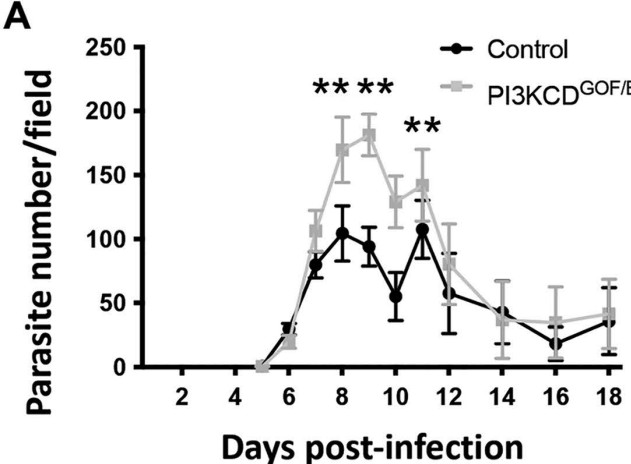

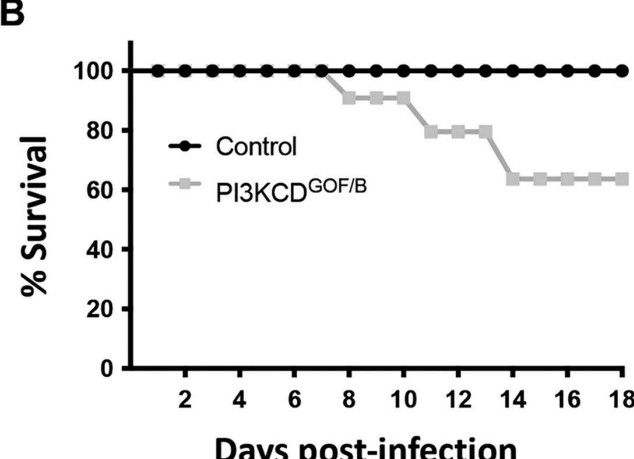

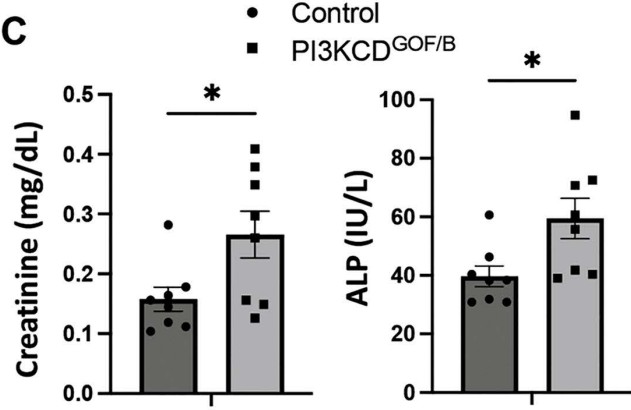

**Fig 1. Impaired control of early *T. congolense* infection in PI3CKD^GOF/B mice.** Groups of PI3KCD^GOF/B and control mice were infected intraperito-neally with 1000 *T. congolense* parasites. (A) Daily parasite counts were estimated using microscopy of tail vein blood smears. (B) Survival curves of infected mice. Results are pooled from 4 independent experiments (n = 26 total per group). At day 7 post-infection, serum levels of (C) Creatinine and ALP were measured. Results are pooled from 2 independent experiments with similar results (n = 8 mice total per group). Error bars, ± SEM (*, $p < 0.05$; **, $P < 0.01$).

consistent with more severe morbidity affecting kidney and liver functions (Fig 1C). These findings suggest that PI3Kδ hyperactivation in B cells impairs the control of *T. congolense* infection, leading to more severe systemic disease progression and reduced host survival.

## PI3KCD^GOF/B mice generate robust antibody responses to *T. congolense* infection

Adequate production of antibodies has previously been associated with the effective control of Trypanosomiasis in mice [34,40,41]. Hence, we determined whether the susceptibility to infection observed in PI3KCD^GOF/B mice was associated with sub-optimal humoral immune responses. Our results indicate that total serum IgM, IgG1 and IgG2a antibodies were elevated on days 4 and 7 and not different from the control group at day 11 (Fig 2A-2C). Trypanosome-specific IgM antibodies were increased at all time points (Fig 2D). Trypanosome-specific IgG1 and IgG2a were similar to controls at days 4 and 7 but reduced at day 11 (Fig 2E/2F). These results indicate that PI3KCD^GOF/B mice generate a robust IgM antibody response and slightly reduced antigen-specific IgG responses by day 11.

## Robust germinal center B cell and plasma cell responses are generated in *T. congolense*-infected PI3KCD^GOF/B mice

We further assessed generation of germinal center (GC) and plasma cell responses in the spleen using flow cytometry (complete gating strategies provided in S1 Fig. GC responses play a crucial role in generating T-dependent humoral immunity, enabling the production of antigen-specific immunoglobulins necessary for parasite clearance. We examined the frequencies of GL7^+Fas^+ GC B cell populations and found that percentages and absolute numbers of GC B cells in the spleen of the PI3KCD^GOF/B mice were not significantly different from controls on days 4 and 7, while absolute numbers were slightly higher on day 11 (Fig 3A). We proceeded to assess CD4^+PD1^+ICOS^+ follicular helper T cells (T_FH) that are critical participants in the GC response and found also no significant differences at days 4 and 7, and a slight elevation in absolute cell numbers at day 11 (Fig 3B). Finally, we examined CD138^+ plasma cells in both groups of mice and observed that CD138^+B220^- plasma cells (Fig 3C) and CD138^+B220^+ plasmablasts (S2A Fig) were similar in both the PI3KCD^GOF/B mice and their littermate controls. However, we noted a significantly increased proportion of IgM^+ plasma cells in PI3KCD-^GOF/B mice, consistent with reduced switching to other antibody isotypes (S2B Fig). Together, these results suggest that the susceptibility to early infection observed with hyperactivation of PI3Kδ in B cells is not associated with impaired germinal center and plasma cell responses during *T. congolense* infection.

## PI3KCD^GOF/B mice exhibit an altered balance of pro- and anti-inflammatory cytokines during *T. congolense* infection

PI3KCD^GOF mutations were previously found to impact the generation of IL10-producing regulatory cells [6,42], which could potentially impact the generation of effective cell-mediated immune responses contributing to parasite control. We proceeded to evaluate the levels of cytokines IFNγ, IL12/IL23p40, TNFα, and IL10 in blood or peritoneal fluid before and during early *T. congolense* infection (Days 0, 4 and 7). The results showed that mutant mice have reduced levels of the pro-inflammatory cytokine IFNγ at day 4 post-infection (Fig 4A), as well as reduced IL12/IL23p40 and TNFα in blood at day 7 post-infection (Fig 4B/4C). In contrast, IL10 levels were significantly increased in both blood and peritoneal fluid prior to infection at day 0, and PI3KCD^GOF/B mutants also mounted stronger IL10 responses at day 4 and 7 post-infection (Fig 4D). Calculation of the IFNγ:IL10 ratio for each mouse revealed a significantly decreased IFNγ:IL10 ratio in the mutant mice (Fig 4E). Consistent with serum and peritoneal wash analyses, flow cytometry analyses revealed elevated overall frequencies of IL10-producing cells in spleen at both baseline and day 4 post-infection, and substantially elevated frequencies of IL10-producing cells in peritoneal cavity at baseline and throughout infection (S3A Fig). In contrast, similar frequencies of IFNγ-producing cells were observed in spleen, whereas the frequency of IFNγ-producing cells was reduced in the peritoneal cavity (S3B Fig). Together, these results indicate that B cell-intrinsic hyperactivation of PI3KCD generated

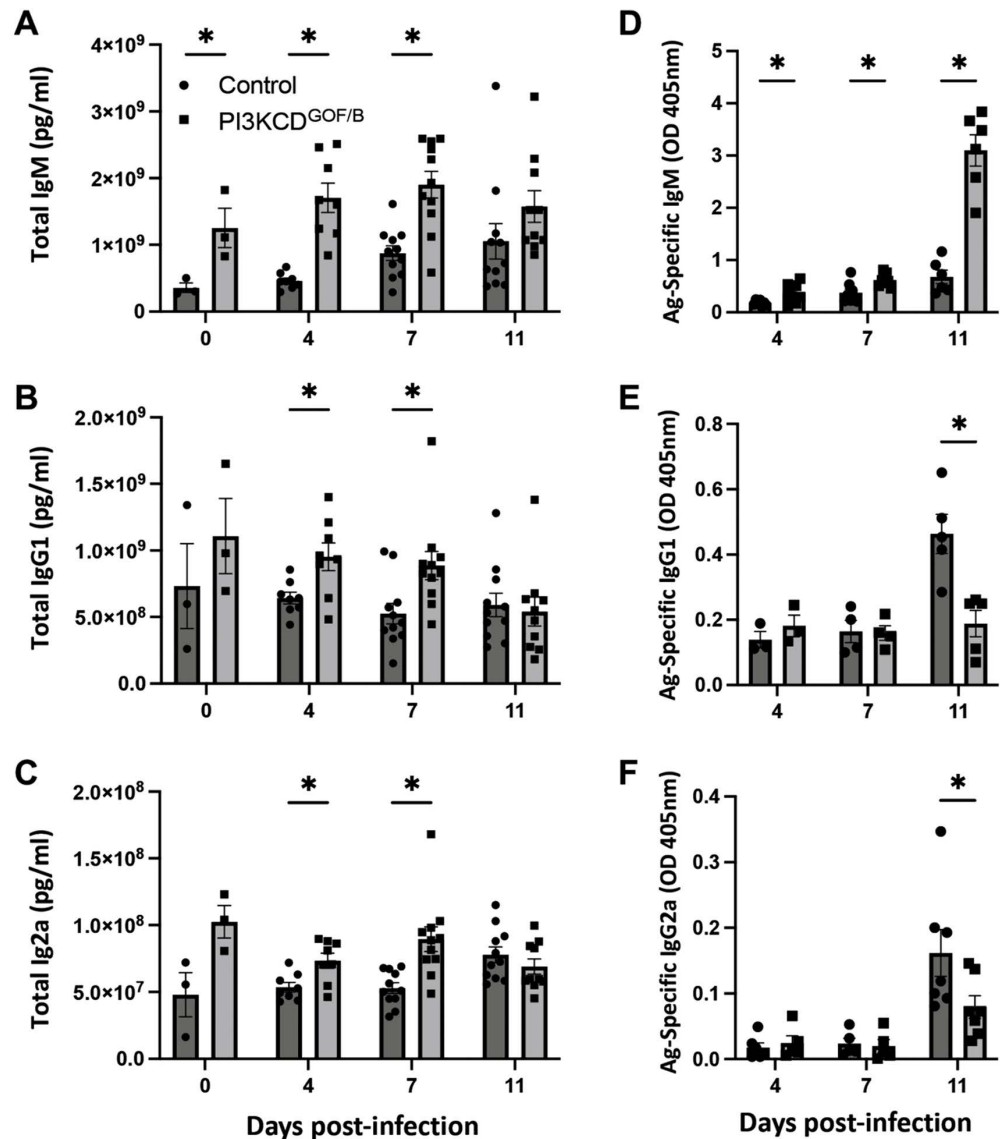

**Fig 2. Impact of B cell-intrinsic PI3CKD hyperactivation on antibody responses.** Serum samples were analyzed by ELISA for total (A) IgM, (B) IgG1 and (C) IgG2a. Antigen-specific (D) IgM, (E) IgG1 and (F) IgG2a were similarly assessed. Results for total antibodies are pooled from 3 independent experiments (n = 8-12 mice per group, per time point). Antigen-specific antibodies are pooled from 2 independent experiments with similar results (n = 4-6 mice per group, per timepoint) Error bars, ± SEM (*, p < 0.05).

a more anti-inflammatory immune milieu upon infection, potentially compromising the effector responses required for disease control.

### Expansion of IL10-producing B cells in PI3KCD^GOF/B mice at baseline and during early infection

Using a genetically encoded IL10-GFP reporter, we examined which cells are producing IL10 in the PI3KCD^GOF/B model. Consistent with previous studies [6], B cells show elevated levels of IL10 expression in PI3KCD^GOF/B mice at baseline and days 4–7 of infection (Fig 5A/5B). We determined absolute numbers of IL10-producing B cells, T cells or non-B/T cells

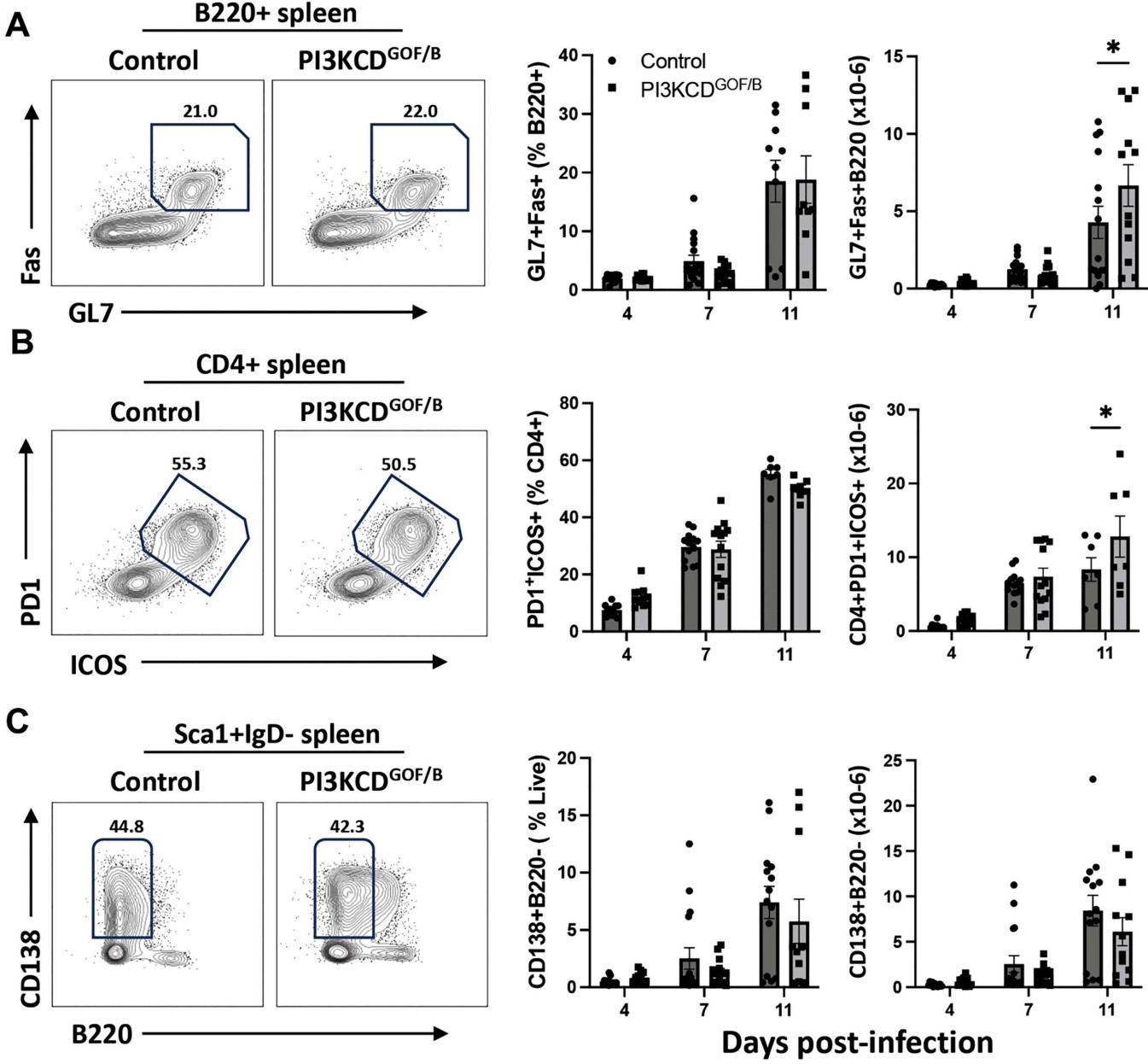

**Fig 3. PI3KCD hyperactivation in B cells does not substantially alter germinal centre and plasmacyte responses during *T. congolense* infection.** Splenocytes were isolated on the indicated days post-infection and analyzed by flow cytometry. (A) GC B cells (B220$^+$GL7$^+$Fas$^+$) and (B) T$_{FH}$ cells (CD4$^+$PD1$^+$ICOS$^+$) were gated as shown in left panels and data expressed as percent of B or CD4+ T lymphocytes (middle panels) or absolute numbers per spleen (right panels). (C) Plasmacytes (CD138$^+$B220$^-$) within Sca1$^+$IgD$^-$ cells were gated as shown in left panels and data expressed as percent of live cells (middle panel) or absolute numbers per spleen (right panel). Results are pooled from 3 independent experiments with similar results. (n = 10–16 mice total per group, per timepoint). Error bars, ±SEM (*, p < 0.05).

and found that IL10$^+$ B cells are increased at baseline and B cells are the predominant IL10-producing cell type in PI3K-CD$^{GOF/B}$ spleen and peritoneal cavity at day 4 post-infection; however, by day 7, increased IL10-producing T cells and non-T/B cells could also be observed in the spleens of mutant mice. In the peritoneal cavity, B cells remain the dominant

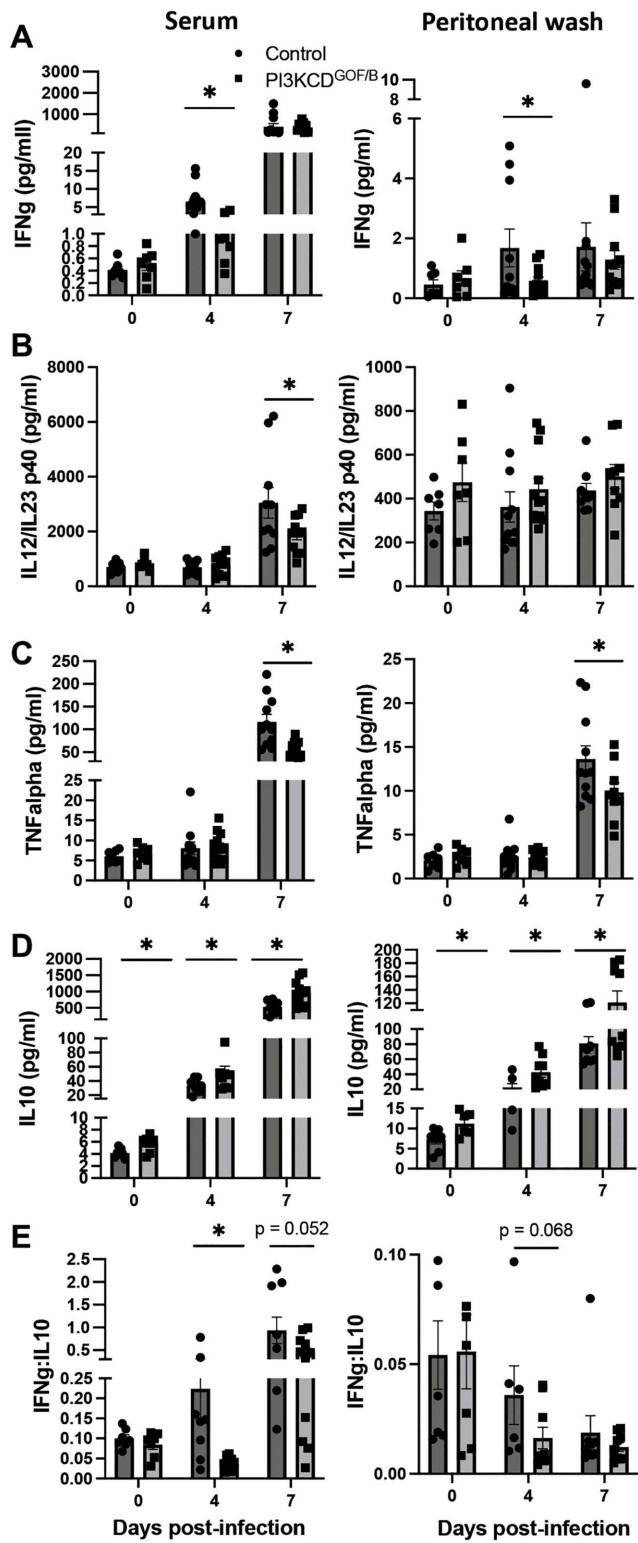

**Fig 4. Cytokine profile of PI3KCD**GOF/B** mice during early *T. congolense* infection.** Serum and peritoneal fluid were analyzed using Mesoscale assay for (A) IFNγ, (B) IL12/IL23p40, (C) TNFα, and (D) IL10 at the indicated time points. (E) IFNγ/IL10 ratios were also determined. Results are pooled from 3 independent experiments with similar results (n = 12 mice per group, per timepoint). Error bars, ± SEM (*, p < 0.05).

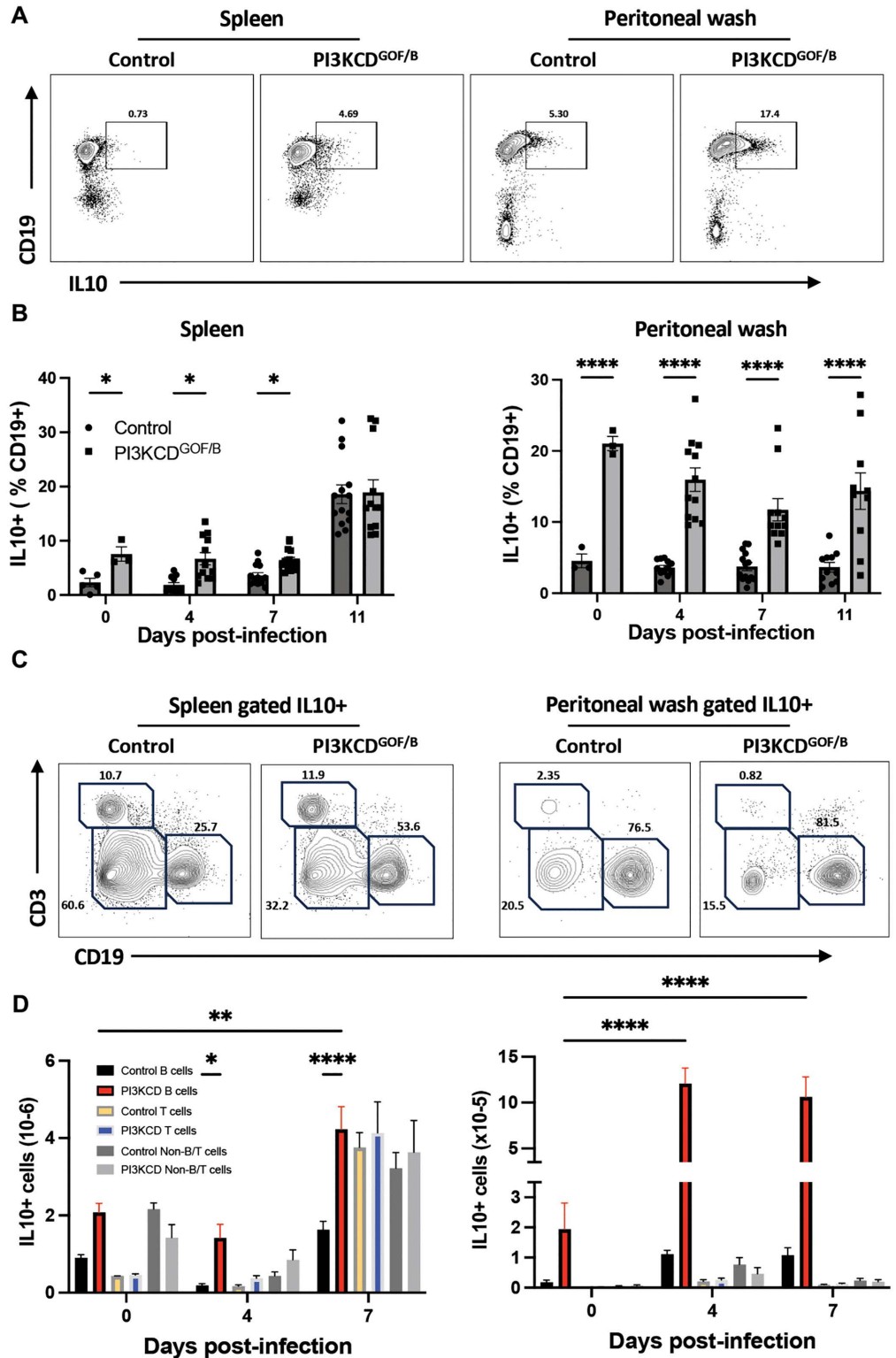

**Fig 5. Expanded IL10+ B cells in PI3KCD**[GOF/B] **mice.** Splenic and peritoneal B cells were isolated, stained and analyzed using flow cytometry. (A) Flow plots of spleen and peritoneal cells showing an increased proportion of CD19 cells expressing the IL10-GFP reporter. (B) Frequency of IL10+ B cells in spleen or peritoneal cavity, at various timepoints pre- and post-infection. (C) Flow plots of the distribution of IL10+ cells in the spleen and peritoneum, (D)

Absolute numbers of IL10-producing B cells, T cells or non-B/T cells in the spleen (left) and peritoneum (right) at days 0,4 or 7 post-infection. Results are pooled from 3 independent experiments with similar results. (n = 12-14 mice per group, per timepoint). Error bars, ± SEM (*, p < 0.05, **, P < 0.01, ****, P < 0.0001).

IL10-producing cell type at day 7. These findings suggest that B cell-intrinsic hyperactivation of PI3Kδ drives the expansion of IL10-producing B cells, generating a more immunosuppressive milieu.

### Phenotype and transcriptomic analysis of expanded IL10+ B cells in PI3KCD<sup>GOF/B</sup> mice

We examined the phenotype of expanded IL10+ B cells in PI3KCD<sup>GOF/B</sup> mice using flow cytometry and RNA sequencing. We observed that most of the expanded IL10+ B cells expressed B1 cell markers CD5 and CD43 (S4A Fig), consistent with previous reports of PI3KCD-driven expansion of innate-like B cells [43] and the known propensity of B1 cells to produce IL10 [44–46]. These cells also exhibited downmodulation of surface IgM, consistent with chronic BCR signaling. IL10- B cells also showed a trend of IgM downmodulation and increased CD5 expression, albeit substantially less than IL10+ B cells (S4B Fig). Following infection, the phenotype of IL10+ B cells was largely maintained in the peritoneum but showed dynamic changes in the spleen, including increased IL10 expression and further IgM downmodulation consistent with B cell activation (S4C Fig).

We FACS sorted IL10+ CD19+ B cell populations from the spleen and peritoneum of both groups of mice and compared their gene expression patterns to IL10- CD19+ B cells from the same mice using bulk RNA sequencing analysis. Principal component analysis revealed that the largest source of variability (PC1, 54%) was related to the tissue source of B cells (i.e., spleen versus peritoneal cavity), and the next largest source of variability (PC2, 31%) was associated with IL10 expression status (Fig 6A/6B). Notably, the IL10-expressing cells from PI3KCD<sup>GOF/B</sup> and littermate controls grouped closely in PCA analysis, indicating that the expanded population in PI3KCD<sup>GOF/B</sup> are relatively similar to normal IL10-producing regulatory B cells transcriptionally. Examination of the top differentially expressed genes between IL10+ and IL10- cells further illustrated the similarity between IL10+ populations from PI3KCD<sup>GOF/B</sup> and control mice, and differences between IL10+ cells isolated from spleen and peritoneal cavity (Fig 6C). However, PC3 representing 6% of variability, was related to differences between PI3KCD<sup>GOF/B</sup> and control cells, indicating that hyperactive PI3KCD can drive some gene expression differences in both IL10+ and IL10- B cells. Gene expression changes driving PC2 and PC3, as well as differential gene expression analyses for PI3KCD<sup>GOF/B</sup> versus control cells, are detailed in S5 Fig.

### Expression of inhibitory molecules in expanded IL10+ B cells of PI3KCD<sup>GOF/B</sup> mice

We investigated the expression of several immune regulatory genes within PI3KCD<sup>GOF/B</sup> and control B cell populations. As expected IL10 mRNA was only significantly expressed in sorted IL10-GFP+ cells, and IL10 mRNA levels within the sorted IL10+ population were about twofold higher in the PI3KCD<sup>GOF/B</sup> group (Fig 7A). Consistent with the findings above, the mean intensity of IL10-GFP reporter expression within the IL10+ B cell population was similarly elevated in the spleen and peritoneum of the PI3KCD<sup>GOF/B</sup> mice (Fig 7B). Additional regulatory genes such as adenosine deaminase [47,48], TIM1 [49], CTLA4 [50] and IL35 subunits [51] were also found to be expressed in IL10+ populations of both PI3KCD<sup>GOF/B</sup> and control mice (S6 Fig), suggesting that expanded IL10+ cells PI3KCD<sup>GOF/B</sup> have regulatory capacity via multiple mechanisms. Immunomodulatory adenosine-metabolizing receptors CD73 and CD39 were examined at the level of mRNA expression and cell surface protein detectable by flow cytometry (Fig 7A/7B). In control mice prior to infection, IL10+ B cells showed higher levels of CD73 and CD39 mRNA, and their corresponding cell surface proteins, compared to IL10 negative B cells. PI3KCD<sup>GOF/B</sup> also showed CD73 and CD39 expression among their expanded IL10+ cells; however, their expression of CD73 was significantly lower than the control IL10+ counterparts (Fig 7A/7B). Post-infection flow cytometry analysis of IL10+ B cells showed that the PI3KCD<sup>GOF/B</sup> group retained elevated mean expression of IL10 but showed

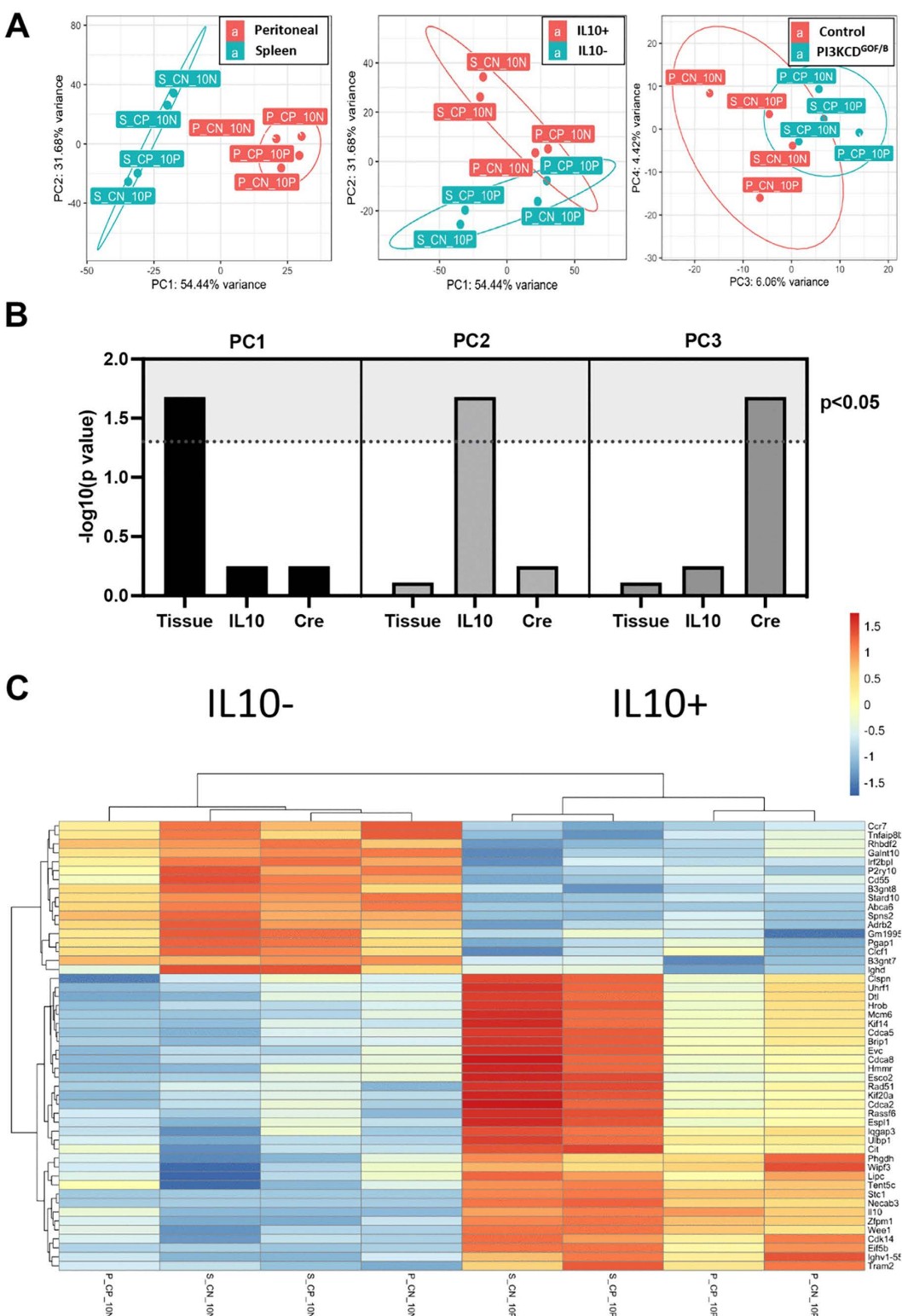

**Fig 6. Transcriptome profiles of IL10-expressing B cells in PI3KCD**[GOF/B] **and control mice.** Mouse splenic and peritoneal B cells were isolated from PI3KCD[GOF/B] mice and littermate controls. The cells were stained and FACS sorted to obtain live CD19+CD4-IL10+ and live CD19+CD4-IL10- cell populations. Gene expression differences were determined using bulk RNA sequencing and analysis. Sample notation: S = spleen, P = peritoneal,

CP = PI3KD^GOF/B, CN = control, 10N = IL10⁻, 10P = IL10⁺. (A) Principal component analysis plots. (B) Relation of principal components (PCs) with sample covariates, showing that PC1 is significantly associated with tissue origin (spleen vs peritoneal), PC2 is significantly associated with IL10 expression, and PC3 is significantly associated with PI3KD^GOF/B mutation. (C) Heatmap depicting the top 50 differentially expressed genes according to adjusted p value, comparing IL10⁺ vs IL10⁻ groups.

reduced CD73 and CD39 levels at day 4 (Fig 7C). From the above results, we can infer that the hyperactivation of PI3Kδ selectively in B cells drives the expansion of increased numbers of IL10-producing regulatory cells that show an overall similarity to normal IL10-expressing regulatory B cells, but also exhibit perturbed expression patterns of some specific inhibitory molecules.

## B cell-intrinsic PI3K hyperactivation alters macrophage and T cell activation and effector function during *T. Congolese* infection

We examined how B cell-intrinsic hyperactivation of PI3Kδ in B cells can impact the function of surrounding immune cells that mediate key immune effector functions, such as T cells and myeloid cells. CD4⁺ T cell numbers in the spleen expanded similarly, with mutants showing a slight increase in T cell numbers at day 11 (Fig 8B). This contrasted with the peritoneum where CD4⁺ T cell numbers were diminished at the peak of infection and showed a significant decrease at day 11 in the mutants (Fig 8B). While the frequency of IFNγ expression among CD4⁺ T cells was similar in mutant and controls at days 4 and 7, a trend towards reduced IFNγ was seen at the day 11 timepoint, reaching significance in the peritoneum (Fig 8C). At day 11, we also noted a trend of reduced frequencies of IL4⁺ and IL17⁺ T cells in the spleen (S7A Fig). Among peritoneal cavity CD4 T cells, a significant reduction in IL17⁺ and IL10⁺ cells was observed (S7B Fig). Macrophage numbers in the spleen were similar between the two groups at the three assessed time points (Fig 8E); however, reduced macrophage expression of CD86 was observed in PI3KCD^GOF/B mice (Fig 8F), suggesting an impairment in macrophage activation. Consistent with this, reduced macrophage expression of inducible nitric oxide synthase (iNOS) was observed (Fig 8G). This observation, together with the previously observed decrease in macrophage-activating cytokines such as TNFα and increased anti-inflammatory IL10, led us to examine the production of nitric oxide (NO), a key effector molecule made by macrophages in targeting *T. congolense*. We assessed levels of nitrite ($NO_2$-), one of the key breakdown products of NO, in supernatants of splenocytes from infected mice restimulated with Trypanosome parasite lysate and observed significantly reduced $NO_2$- generated from PI3KCD^GOF/B mice (Fig 8H). The above results indicate that B cell-specific PI3Kδ hyperactivation impairs macrophage and T cell activation and effector function.

To determine whether blocking excess IL10 can correct any of the abnormal immune responses in PI3KCD^GOF/B mice, we treated mice with a neutralizing antibody against IL10. Several aspects of the immune response were impacted, including increased antigen-specific antibody, trends of increased plasma cells, IFNγ -producing T cells in spleen and increased nitric oxide production by splenocytes (Fig 9A-9D). However, anti-IL10 treatment did not significantly alter markers of macrophage activation or parasitemia (S8 Fig), indicating that blockade of a single factor produced by expanded Bregs is not sufficient to fully restore immune defence.

## Discussion

Our results show that B cell-intrinsic PI3KCD hyperactivation leads to increased susceptibility to *T. congolense* in early infection through changes in the immune regulatory milieu. We found the presence of an abnormally expanded IL10-producing B cell population in both the spleen and peritoneal cavity of PI3KCD^GOF/B mice, which are equipped with multiple inhibitory molecules that could impair the generation of protective responses needed to control the first wave of parasitemia. The dominant cytokine responses in the early phase of infection are critical for shaping the subsequent immune response [52], as has been reported in infectious diseases such as COVID-19, malaria and leishmaniasis where the

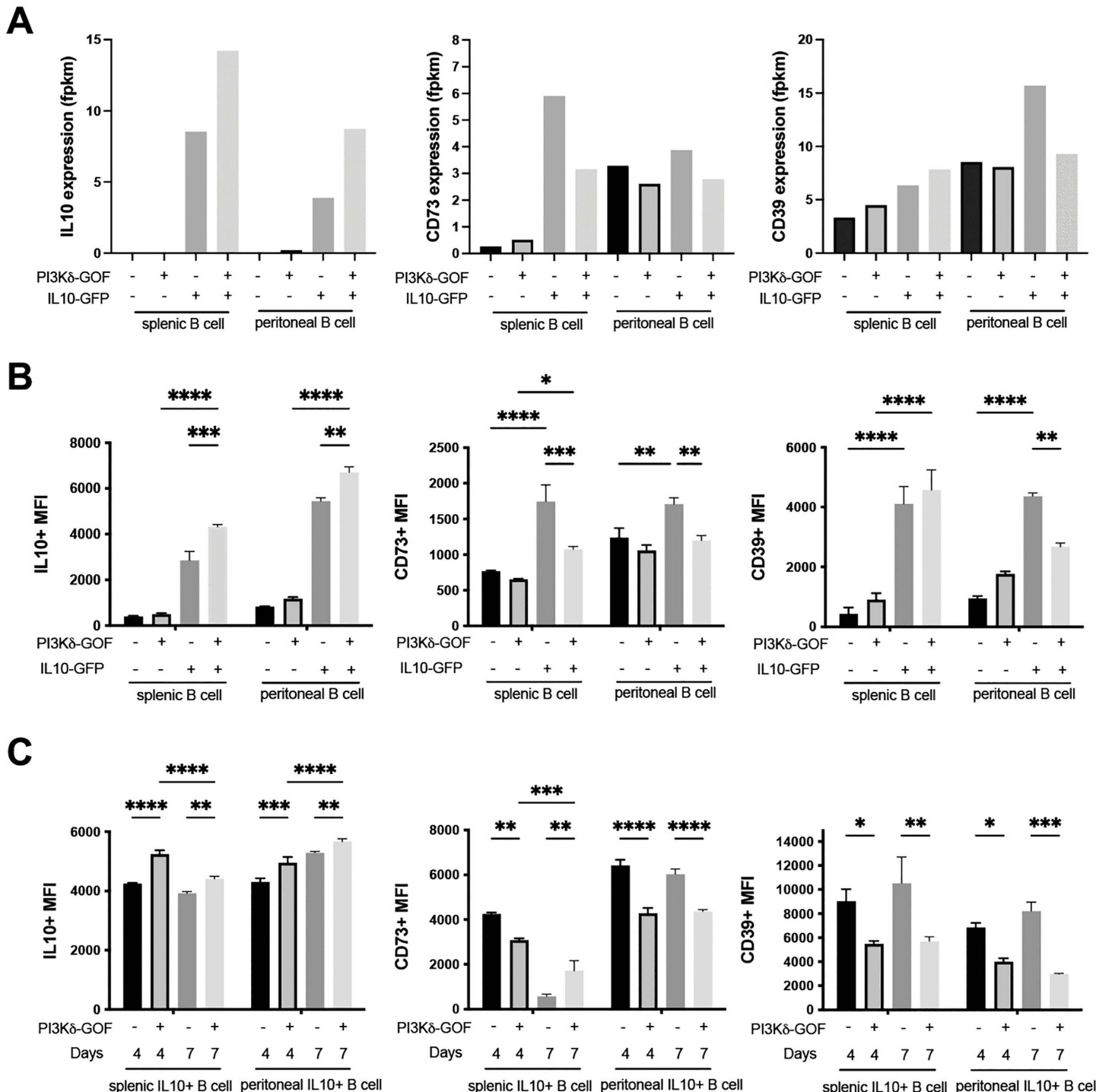

**Fig 7. Inhibitory molecule expression within IL10-expressing B cells in PI3KCD[GOF/B] and control mice.** (A) mRNA expression of inhibitory molecules IL10, CD73 and CD39 as determined from RNA sequencing data. Protein expression of IL10, CD73 and CD39 was determined by flow cytometry pre-infection (B) and post-infection (C). Error bars, ±SEM (*, p < 0.05; **, P < 0.01, ***, P < 0.001, ****, P < 0.0001).

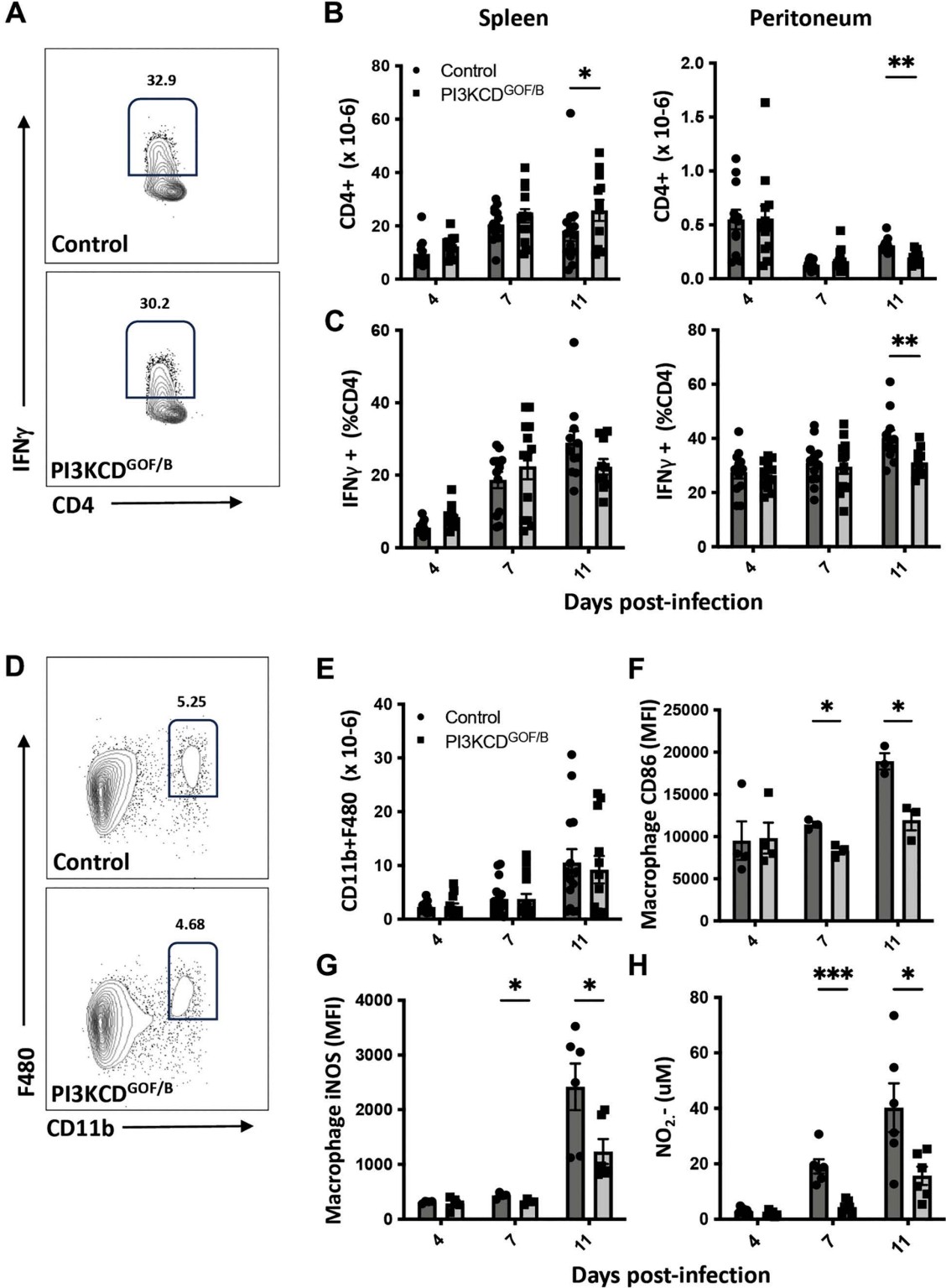

**Fig 8. Impact of B cell-intrinsic PI3CKD hyperactivation on T cell and macrophage activation in *T. congolense* infection.** Spleen and peritoneal cavity cells were isolated and assessed by flow cytometry on the indicated time points post-infection. (A) Flow plots of the spleen at day 11 showing IFNγ⁺ T cells, (B) CD4⁺ T cell numbers, (C) CD4⁺ T cell IFNγ production. Flow plots of the spleen at day 11 showing (D) CD11B⁺ F480⁺ Macrophage cells,

(E) Macrophage absolute numbers, (F) Macrophage CD86 expression or (G) macrophage iNOS expression. (H) Nitrite levels in spleen cell cultures stimulated with TC13 lysate were measured using the Griess assay on culture supernatants. CD4$^+$ T and Macrophage cell results are pooled from 3 independent experiments with similar results (n = 12-16 mice per group, per timepoint). Flow cytometry data are representative of 3 independent experiments with similar results (n = 3-6 mice per group, per timepoint). NO$_2$- results are pooled from 2 independent experiments with similar results (n = 6 mice per group, per timepoint). Error bars, ± SEM (*, p < 0.05).

presence of an early dominant pro-inflammatory cytokine response by IFNγ is correlated with the decrease in susceptibility to infection [53–55]. This is also the case in trypanosomiasis, as production of the cytokine IFNγ from T cells and natural killer (NK) cells is required to drive effector responses such as the development of T helper 1 cells and macrophage activation [33,56]. We observed that B cell-intrinsic PI3KCD hyperactivation alters the baseline host immune environment with expanded immune-suppressive B cell populations. The presence of these cells and the inhibitory molecules they employ may tilt the immune environment towards an anti-inflammatory state putting the immune system at a disadvantage. These expanded populations may be functionally equivalent to the IL10-producing B cell populations found to be elevated in PBMC's from the blood of APDS patients [6,57], and which contribute to immune dysfunction in APDS.

Our study shows that B cell-intrinsic PI3KCD hyperactivation can compromise macrophage activity and function. Macrophages are key players in trypanocidal activity during infection. They express iNOS and secrete the parasite-targeting molecule nitric oxide [58,59]. We observed that both iNOS and NO were significantly impaired in macrophages from mutants, potentially due to the influence of the more immune-suppressive environment generated by expanded IL10-producing B cells. These findings are consistent with Suchanek and colleagues [60] and a few other groups who have suggested a role for tissue-resident innate-like B cell populations in regulating macrophage polarization and functions, partly via IL10-dependent mechanisms [61–63]. Although our study did not directly assess cytokine production by macrophages, the blood cytokine levels in the mutant group revealed a marked reduction in two key cytokines produced by macrophages during infection, TNFα and IL12/IL23p40 [64], which could potentially be contributing to the compromised host response observed early in the disease. We additionally observed evidence that CD4 T cell functional responses were compromised, with significantly reduced frequencies of IFNγ and IL17 producing cells in the peritoneal cavity. These altered T cell responses may reflect impaired innate responses or direct effects of regulatory B cells on T cell activation. The latter possibility would fit with the larger expansion of regulatory B cells within peritoneal cavity and correspondingly greater impact on T cell responses in this location.

Antibody responses are also an essential component of anti-trypanosome immunity, and it is possible that PI3KCD hyperactivation could impact this aspect of the response. Previous studies have shown that B cell-intrinsic PI3KCD hyperactivation can impair the generation of antigen-specific isotype-switched antibodies after immunization with model antigens [4]. This could be attributed partly to effects on isotype switching and partly to effects on plasma cell maturation and survival [65]. Previous studies have shown that Trypanosoma infection causes polyclonal B cell activation and disruption of splenic architecture, making it difficult to histologically distinguish classical germinal center versus extrafollicular responses [40,66]. We observed the appearance of GC and T$_{FH}$ phenotype cells as well as robust plasmablast responses upon infection, neither of which were obviously different in mutant mice. We also observed reduced antigen-specific IgG responses at day 11, however, very robust IgM responses were generated, and absolute numbers of plasma cells were comparable to controls. It thus appears that the rapid and robust plasmablast response generated by *T. congolense* infection is not markedly impaired by PI3KCD hyperactivation; however, it remains possible that plasma cells numbers might more rapidly decline over time in mutant mice.

We found that antigen-specific IgG2a and IgG1, which contribute to opsonization/complement fixation and effector functions respectively, were slightly reduced after the first week of infection, which could potentially have effects on parasite control in the longer term; however, we did not observe obvious differences in parasitemia among mice that survived past day 14. The hyper-IgM and reduced isotype switch phenotype we observe is in line with what has previously been

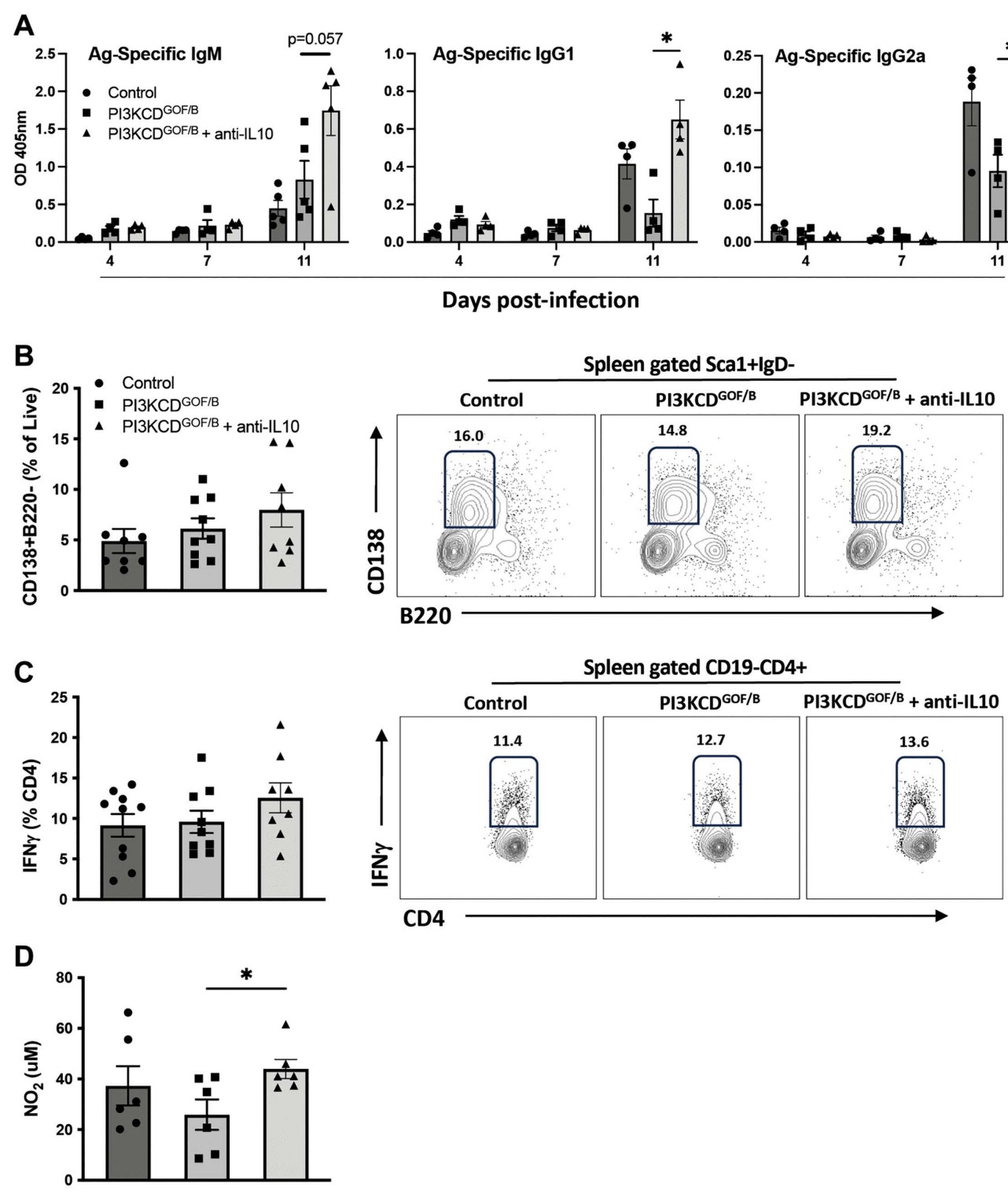

**Fig 9. IL10 blocking in PI3KCD**GOF/B **mice.** Mice were infected with *T. congolense* parasites with or without administration of anti-IL10 blocking antibody. (A) Serum samples were analyzed at the indicated days post-infection by ELISA for antigen-specific IgM, IgG1 and IgG2a. Splenic (B) CD138+B220- plasmacytes and (C) IFNγ+ CD4+ T cells were assessed using flow cytometry. (D) Nitrite levels produced in spleen cell cultures stimulated with TC13

lysate was assessed using the Griess assay on supernatants. Flow cytometry results are pooled from 2 independent experiments with similar results (n = 8-10 mice per group, per timepoint). Error bars, ±SEM (*, p < 0.05).

documented in human patient cohorts and mouse studies depicting activated PI3KCD syndrome (APDS) [1,4,67–69]. Since this defect appears in mutants only after the first wave of parasitemia has peaked, it seems unlikely that it could account for the early impairment in parasite control. Consistent with this interpretation, IL10 blockade restored class-switched antibody production (measured at day 11), but this was not sufficient to substantially improve control of the first wave of parasitemia (peaking around day 7). However, it remains possible that altered quality and type of antibody response in PI3KCD^GOF/B mice may influence some other aspects of the immune responses, for example, via Fc receptor-mediated macrophage activation.

The expanded IL10-positive B cells in the mutant mice appeared to be related to the normal IL10-producing B1 cell population, but also exhibited some alterations in their gene expression profile. They exhibit increased IL10 mRNA and protein levels and modulated levels of some other unique inhibitory molecules, such as the CD39 and CD73 ectonucleotidases. These enzymes modulate purinergic immune responses by hydrolyzing extracellular "danger molecules," such as ATP and ADP (via CD39), into AMP, which is then converted by CD73 into the immunosuppressive molecule adenosine. This cascade fine-tunes immune activity, promoting an optimized immune regulatory response that is particularly beneficial during acute inflammation [70]. At the mRNA level, we also noted a trend of increased IL35 subunit expression in both spleen and peritoneal IL10^+ B cells from PI3KCD^GOF/B mice, whereas other inhibitory molecules such as adenosine deaminase, TIM1 or CTLA4 were expressed at similar or slightly lower levels. Flow cytometry analysis indicated some changes in the phenotype of IL10^+ B cells in the spleen after infection, which likely reflects both activation of pre-existing IL10^+ cells and induction of new IL10^+ B cell populations. The elevated numbers of regulatory B cells, together with their subtly different expression of inhibitory molecules, may be contributing to the altered immune regulatory milieu in PI3KCD^GOF/B mice before and during infection.

Since IL10 production is one of the hallmarks of regulatory B cells we examined whether blockade of this factor could correct the abnormal responses to Trypanosome infection in PI3KCD^GOF/B mice. The results show that IL10 blockade did impact some aspects of the immune response including B cell, T cell and macrophage responses, but this was not sufficient to normalize parasitemia. These findings suggest that expanded IL10-producing B cells likely utilize additional immunosuppressive mechanisms beyond IL10 itself to regulate immune responses to Trypanosome infection. This is consistent with our results showing that IL10-producing B cells express multiple regulatory factors, and consistent with other studies showing that blockade of a single factor is not sufficient to neutralize Breg functional activity [6]. For example, Stark and colleagues demonstrated in a similar PI3Kdelta gain-of-function model that IL10 blockade was insufficient to reduce disease susceptibility in *Streptococcus pneumoniae* lung infection [6]. Notably, we and others have previously observed disease resistance in mice entirely lacking functional innate-like IL10-producing B cells, such as the PI3Kdelta-loss of function mice [6,34,35]. Beyond the direct regulatory effects of Bregs during infection, increased baseline IL10 exposure may imprint long-lasting functional defects in innate cells such as macrophages, dendritic cells, and even B cells, which make them remain epigenetically or transcriptionally reprogrammed toward a regulatory phenotype, leading to, for example reduced Fcγ receptor and complement activity [71].

One remaining question from our study is the determinants of the observed decreased survival in *T. congolense* infection. Our results indicate only a 30% mortality in the mutant group, with others surviving comparably to the control until later time points. The ability to produce innate and T-dependent antibodies has previously been documented as critical for the control of *T. congolense* infection [34,59,72,73], but with relatively robust antibody production in early infection in the mutant group in this study, we can infer that decreased survival is not likely related to insufficient antibodies. Given the reduction in several proinflammatory cytokines, it also seems unlikely that mortality relates to a "cytokine storm". It is possible that uncontrolled parasite growth itself, or together with the hyper-IgM response and immune complex-mediated damage, leads to lethal tissue pathology in some animals (for example, disrupted liver and renal function).

In conclusion, we have found a significant role for B cell-intrinsic PI3KCD hyperactivation in host immune response to protozoan infection via influence on cytokine profiles, antibody responses and macrophage activation. These results concur with recent studies showing that PI3KCD can drive the expansion of regulatory B cells that profoundly affect the programming of macrophages and response to infection [60].

Beyond APDS, a greater understanding of the mechanisms by which PI3KCD-driven regulatory B cells impact immune function is likely relevant in multiple infectious disease contexts, as well as chronic diseases involving hyperactivation of the PI3KD pathway, such as B cell malignancies and autoimmunity.

## Materials and methods

### Ethics statement

All animal procedures and experiments were reviewed and approved by the University of Manitoba Animal Care Committee (ACC) under protocol number B2021-028/1/2.

### Mice

Mice with a conditional mutant PI3KCD-E1020K allele [4] and an Mb1-Cre allele [74] were generated at the Seattle Children's Research Institute. IL10-GFP reporter mice (B6(Cg)-*Il10*^*tm1.1Karp*/J) [26] were obtained from The Jackson Laboratory (https://www.jax.org/strain/014530). Female PI3KCD^E1021K x Mb1-Cre x IL10-GFP reporter mice heterozygous for the E1020K mutation and their Cre⁻negative littermate controls were used in the study. They were housed in a specific pathogen-free facility at the University of Manitoba, according to the Canadian Council on Animal Care guidelines and used for experiments between 8–12 weeks of age.

### Parasite infection

*Trypanosoma congolense* (Trans Mara strain, variant antigen type TC13) was used for all experiments, and the origin of the strain has previously been reported [75]. To prepare parasites for infection experiments, CD1 mice were immunosuppressed with Cyclophosphamide (Cytoxan; 200mg/kg) intraperitoneally (i.p) and infected i.p. 2 days later with thawed TC13 stabilates [75]. After 3 days, mice were anesthetized with isoflurane and blood containing live parasites was collected via cardiac puncture. Parasites were purified by filtering the blood through a diethylamino ethyl (DEAE) cellulose anion exchange chromatography and then washing in Tris-saline glucose (TSG) solution. The parasites were counted using the hemocytometer and further resuspended in TSG + 10% FBS to the desired concentration ($10^4$/ml). Experimental mice were infected by injecting 100 µl of suspension containing $10^3$ parasites.

### Estimation of parasite burden and endpoint sample collection

Parasite load was estimated by transferring a drop of blood from the tail vein of the mice onto a glass slide (Fisher Scientific Ottawa, ON) and counting the number of blood form trypomastigotes present in at least 12 fields at 400x magnification on the light microscope [76]. At the indicated times after infection, mice were sacrificed, and splenic tissues were obtained. Blood from the saphenous vein or heart (endpoint) and peritoneal aspirates were also collected. Serum was obtained from blood clotted at room temperature for 1 hour and spun in the centrifuge for 10 minutes at 3000RPM. Peritoneal aspirates were collected by injecting 200 µL of PBS into the peritoneum, retrieving the aspirate, and then centrifuging and storing the supernatant at -80°C for further analysis.

### Flow cytometry

Single cell suspensions of spleens were lysed using ACK lysis buffer (0.15 M $NH_4Cl$, 10 mM $KHCO_3$, 0.1 mM Na EDTA) and the cells subsequently washed with PBS and resuspended in RPMI (+2.05 mM L-Glutamine) at a final concentration

of 2 x 10$^6$/ml. Peritoneal cell pellets were resuspended in RPMI at a final concentration of 5 x 10$^5$/ml. Germinal center B cell responses were assessed by staining the spleen cells ex vivo with the following fluorochrome-conjugated antibodies: B220 PerCP (clone:RA3-6B2), CD19 AF700 (clone:1D3/CD19), GL7 APC (clone:GL7), CD95 (clone:SA367H8). Breg characterization was done using CD5 APC (clone:53-7.3),CD43 PE (clone:S11), CD39 PE/Dazzle 594 (clone:Duha59), CD73 PerCP/Cy5.5 (clone:TY/11.8), IgM BV605 (clone:RMM-1), and plasma cells assessed with the markers Sca-1 BV421 (clone:D7), CD138 PE (clone:281–2), IgD AF647 (clone:11-26c.2a) and IgG1 PE/Dazzle 594 (clone:RMG1–1). Antibodies against CD4 BV605 (clone: RM4–5), CD3 AF700 (clone:17A2), ICOS APC (clone: 7E.17G9) and PD1 PE-Cy7 (clone:RMP1–30) were used to identify follicular helper T cells and Macrophages were evaluated using the markers CD11B BV785 (clone:M1/70), F480 PerCP/Cy5.5 (clone:BM8), CD86 PE(clone:GL-1) and iNOS PE-Cy7 (clone: W16030C). All antibodies were obtained from Biolegend. The genetically-encoded GFP reporter described above was used to identify IL10-expressing cells [26]. For intracellular cytokine staining, cells were stimulated with ionomycin (500 ng/mL), phorbol myristic acetate (PMA, 50 ng/mL) and brefeldin A (BFA, 10 μg/mL) for 4hrs, fixed, surface stained for CD4, and subsequently stained for intracellular IFNγ PerCP/Cy5.5 (clone:XMG1.2), IL4 APC (clone:11B11) and IL17 BV421 (clone:TC11-18H10.1) All reagents and antibodies were obtained from Biolegend. After washing with FACS buffer, cells were acquired using the BD FACS Canto II cytometer (BD Bioscience, San Diego CA) and analyzed with FlowJo software (BD Bioscience). The gating strategies used are provided in S1 Fig.

### Bulk RNA-sequencing

Mouse Splenic and Peritoneal B cells from 6-10 mice per genotype were isolated by magnetic bead negative selection using EasySep mouse B cell isolation kit (Stem Cell Technology cat #19854). Cells were pooled and resuspended in FACS buffer, stained with anti-CD19, anti-CD4 and live-dead viability stain as described above, and CD19$^+$CD4$^-$Live-dead$^-$ IL10GFP$^-$ or IL10GFP$^+$ B cells were sorted using a BD FACSAria III Cell Sorter. Cell pellets were processed by Novogene (Sacramento, CA) for bulk RNA sequencing, producing 50–100 million reads per sample. Results were analyzed using R (4.2.0), the DESeq2 (1.36.0) and the pcaExplorer (2.22.0).

### ELISA, Mesoscale and serum biochemistry assays

Total antibody levels for IgM and IgG1 and IgG2a were measured in serum by Enzyme-linked Immunosorbent Assay (ELISA) as previously described [77]. Trypanosome-specific IgM, IgG1 and IgG2a levels were measured by ELISA as previously described [73]. Levels of IL10, IFNγ, IL12/IL23p40 and TNFα in the serum and supernatant from the peritoneal cavity were measured by Mesoscale Discovery U-PLEX assay and performed according to the manufacturer's instructions.

Serum Biochemistry analysis for Creatinine and Alkaline phosphatase (ALP) from serum was carried out at the Centre for Phenogenomics (Toronto ON).

### IL10 Blocking studies

Female PI3KCD$^{E1021K}$ x Mb1-Cre x IL10-GFP mice, heterozygous for the E1020K mutation, were injected intraperitoneally (i.p.) on days 0, 2, 5 and 7 post-infection with 175 μg of purified anti-IL10 antibody (InVivoMAb anti-mouse IL-10, clone: JES5-2A5) each in 200μl of PBS as previously described [78]. Control groups received an equivalent volume of either an isotype-matched control antibody (InVivoMAb rat IgG1 isotype control, clone HRPN) or PBS alone. All mice were monitored daily for parasitemia, mortality and splenic and peritoneal cells assessed by flow cytometry.

### Nitrite assessment

Spleen cells from infected mice were resuspended in complete media (DMEM supplemented with 10% heat-inactivated fetal bovine serum, 100U/mL Penicillin and 100μg/ml streptomycin), at a concentration of 4 x 10$^6$/ml, and transferred into 24 well tissue culture plate (Falcon: VWR Edmonton, AB, Canada) where they were cultured with *T. congolense* lysate (10$^5$ lysate/well) for 72hrs. Nitrite was assessed using the standard Griess reaction [79].

## Statistics

Data are presented as means and standard error of mean (SEM). Student's t-test were used for comparisons of means and SEM between two groups, and non-parametric analysis of variance (one-way or two-way ANOVA) were used to compare means and standard deviations of groups greater than two. Data were analysed using the GraphPad Prism program (GraphPad Software Inc., CA, USA). The error bars are indicative of ± SEM, and the differences are taken as significant at $p < 0.05$. The minimal data set for this study is available in the supplementary data.

## Supporting information

**S1 Fig. Flow gating strategy.** (A) Spleen B, T & Macrophage cell populations, (B) Spleen total frequencies of IFNγ-producing & IL10-producing cells. Bottom right shows subsetting of IL10+ cells to calculate absolute numbers of IL10-producing B, T and non-B/T cells. (C) Peritoneal IL10-producing B cells.
(TIF)

**S2 Fig. Plasma cells in PI3KCD[GOF/B] mice.** Frequencies of (A) Plasmablasts and (B) IgM+ versus IgM- plasma cells in the spleen at the indicated days post-infection.
(TIF)

**S3 Fig. Cytokine analysis from Live cells by flow cytometry.** Frequencies of (A) IL10+ or (B) IFNγ+ cells within spleen (left) or peritoneal cavity (right) at the indicated days post-infection.
(TIF)

**S4 Fig. Histogram overlays and mean fluorescence intensity plots.** Phenotype of (A) IL10-positive B cells at baseline (gated live CD19+IL10GFP+ cells), (B) IL10-negative B cells at baseline (gated live CD19+IL10GFP- cells), and (C) IL10-positive B cells at infection (gated live CD19+IL10GFP+ cells), from control or PI3KD[GOF/B] spleen or peritoneal cavity were analyzed for levels of IL10GFP, CD5, CD43 or IgM expression.
(TIF)

**S5 Fig. Additional analyses of RNAseq data.** (A) Top genes driving variance for PC2 and PC3. (B) Heatmap depicting the top 50 differentially expressed genes according to adjusted p value, comparing PI3KCD[GOF/B] versus control groups.
(TIF)

**S6 Fig. mRNA expression level of selected immune regulatory genes.** Determined from RNAseq data.
(TIF)

**S7 Fig. T cell profile in PI3KCD[GOF/B] mice following infection.** Frequencies of (A) splenic and (B) peritoneal IL4+, IL17+ or IL10+ CD4+ T cells were assessed and expressed as percentages and absolute numbers at the indicated day post-infection.
(TIF)

**S8 Fig. Anti-IL-10 treatment did not significantly affect macrophage activation or parasitemia levels.** Splenic (A) Macrophage frequencies by flow cytometry, (B) CD86 expression and (C) iNOS expression at day 11 post-infection. (D) Parasitemia and survival curves following IL10 blocking in PI3KD[GOF/B] mice.
(TIF)

**S9 Fig. Minimal data file containing key figure datasets.**
(XLSX)

## Acknowledgments

The Authors acknowledge Dr. Christine Zhang, head of the Flow core facility for her expertise in the flow panel setup and contribution to data analysis.

## Author contributions

**Conceptualization:** Folayemi Olayinka Adefemi, Aaron J. Marshall.

**Data curation:** Folayemi Olayinka Adefemi.

**Formal analysis:** Folayemi Olayinka Adefemi, Xun Wu, Milad Sabzevary-Ghahfarokhi.

**Funding acquisition:** Aaron J. Marshall.

**Investigation:** Sen Hou, Jude Uzonna.

**Methodology:** Folayemi Olayinka Adefemi, Xun Wu, Sen Hou, Michelle N. Wray-Dutra, David J Rawlings, Jude Uzonna.

**Project administration:** Aaron J. Marshall.

**Resources:** Michelle N. Wray-Dutra, David J Rawlings, Jude Uzonna, Aaron J. Marshall.

**Software:** Xun Wu, Milad Sabzevary-Ghahfarokhi.

**Supervision:** Aaron J. Marshall.

**Validation:** Folayemi Olayinka Adefemi, Sen Hou, David J Rawlings.

**Visualization:** Aaron J. Marshall.

**Writing – original draft:** Folayemi Olayinka Adefemi.

**Writing – review & editing:** Folayemi Olayinka Adefemi, Xun Wu, Sen Hou, Milad Sabzevary-Ghahfarokhi, Michelle N. Wray-Dutra, David J Rawlings, Jude Uzonna, Aaron J. Marshall.

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
