## [Decision Letter · Decision Letter 0]

12 Nov 2024

Response to Reviewers
Revised Manuscript with Track Changes
Manuscript

Editor-in-Chief

PLOS Pathogens

**Additional Editor Comments (if provided):**
**Journal Requirements:**
**Reviewers' Comments:**

**Part I - Summary**

Reviewer #1: PI3KCD (A subunit of class I PI3K) is a critical enzyme for B cell development and antibody production. PI3KCD gain of function (GOF) mutation is associated with Activated PI3KCD Syndrome (APDS) and is prone to repeated infection. Olayinka-Adefemi et al., determine whether B cell-specific GOF mutation (E1021K) impacts the immune response to the protozoan parasite T. congolense. The authors find that GOF mutant developed a significantly higher parasite burden during the peak of infection between days 7-11 and exhibited approximately 30% mortality during the first two weeks post-infection. GOF (E1021K) infected mice have increased Trypanosome-specific IgM but reduced gG1 and IgG2a, and GOF infected mice form an increased number of GC B cells and Tfh cells, and a similar number of plasma cells/ plasmablast. The authors claim that IL10-producing B cells expanded in GOF mutant mice during early infection, and the ratio of pro-inflammatory to ant-inflammatory is reduced, which results in a higher anti-inflammatory response upon infection. IL10-producing B cells perturb the expression of inhibitory molecules (such as CD73) and impair macrophage activation and effector function in the GOF mutant mice upon infection. The study well-characterized phenotypes associated with parasite burden and mortality. This study didn’t address whether an increase in IL10 or IL-producing B cells is causing the disease or is an outcome of the disease.

Reviewer #2: This manuscript is a straight-forward paper showing the effects of activated PI3Kd expression in B cells on murine responses to Trypanosome congolense. As has been seen in other infection models (Stark et al 2018), an expanded population of IL-10-producing B cells is associated with increased mortality. The characterization is solid, but does not address whether this increased mortality or altered responses result from this population.

Reviewer #3: Dr. Olayinka-Adefemi et al. present an intriguing study on the impact of dysregulated PI3KCD activity on regulatory B cell function, specifically in the context of immune responses to a systemic protozoan parasite infection. They demonstrate that PI3KCD hyperactivation in PI3KCDE1021K/B mice increases susceptibility to infection, linked to a shift in B cell activation towards a regulatory phenotype, marked by an increase in IL-10-producing B cells. The study identifies a molecular program akin to innate-like regulatory B cells and highlights expression of multiple inhibitory molecules in these cells. Although some aspects of the manuscript could benefit from further refinement, the study addresses an important question and offers significant insights for the field.

**Part II – Major Issues: Key Experiments Required for Acceptance**

Reviewer #1: 1. IL-10-producing Breg can either be beneficial or pathological, depending on the context. I think it is important to show the function of Breg (either by depleting Breg or IL-10) in the context of B cell-specific GOF mutant mice and T. congolense infection.

2. To explain the generation of altered antibody isotypes, I think it is important to look at the proportion and number of cells expressing IgM and IgG by surface staining of GC B cells and intracellular staining for plasma blast/plasma cells.

3. Proportion IL10 producing B cells remain high at day 0 and have not increased during Trypanosome-infection (Fig 5). In fact, the proportion of IL10-producing B cells is decreased in the peritoneal cavity of GOF mice after infection. The number of IL-10-producing B cells remains high at day 4 post-infection. The author needs to show IL-10-producing B cell numbers at day 0 to support the claim that trypanosome infection generates higher IL-10-producing B cells.

Reviewer #2: 1,The authors conclude that “that B cell-intrinsic PI3KCD hyperactivation can compromise macrophage activity and function”. Although the authors have previously characterized the B cell-intrinsic PI3KE1020K/b mice (Wray-Dutra et al), it would be useful to confirm that the mutant is not expressed in the macrophages after infection, as a control.

2. What happens to responses if mixed bone marrow chimeras are infected? Is this a dominant effect? Ultimately, is increased IL-10 responsible for these phenotypes? Could the authors use a titration of blocking anti-IL-10 to address this to see if there is a dose where death is reduced in the mutant mice? Some additional characterization would be helpful.

3. It would also be useful to see what happens in mice that constitutively express the activated mutant in all cells (that express PI3Kd).

In summary, this is a solid paper that describes the phenotype of mice expressing an activated PI3K allele in B cells in response to T. congolense. It complements their previous study in kinase-inactive PI3Kd mice and studies on IL-10-expressing B cells. It would benefit from some extra evaluation of these populations, as well as extra data (data points, representative flow plots) to support their summary graphs.

Reviewer #3: Major Comments:

1. The authors report that PI3KCD hyperactivation diminishes antigen-specific IgG responses by day 11 while inducing a robust IgM response, both associated with higher parasite burdens. Although previous studies suggest PI3KCD hyperactivation impairs the development of antigen-specific antibodies, the role of these antibodies in controlling or sustaining parasite burden remains unclear. An adoptive transfer experiment using antibodies from infected WT and PI3KCDE1021K/B mice would help determine their role in modulating infection outcomes.

2. A notable expansion of IL-10-producing B cells is associated with increased parasite burden and mortality in PI3KCDE1021K/B mice; however, the direct role of IL-10 in this context is not fully elucidated. Blocking the IL-10 receptor in T. congolense-infected PI3KCDE1021K/B mice could provide mechanistic insights into IL-10’s role in immune regulation and its contribution to increased susceptibility associated with PI3KCD hyperactivation.

3. The study shows that PI3KCD hyperactivation alters effector memory CD4 T cell differentiation, though no significant differences in IFNγ+ CD4 T cells were observed in infected PI3KCDE1021K/B mice. It would be valuable to also examine Th2, Treg, and Th17 populations, as these subsets contribute regulatory cytokines like IL-10 that might further elucidate the observed immune dysregulation.

**Part III – Minor Issues: Editorial and Data Presentation Modifications**

Reviewer #1: Figure quality is poor, which is very prominent in Figure 6. Figures 6 and 7 are hard to read.

Reviewer #2: The authors show summary graphs for most of the data in the paper. It would be helpful if they showed individual data points. The authors show a representative gating strategy in Figure S1, but it would be useful to include the flow plots for the measured parameters in the main figures.

In Fig S4 it would be helpful to show levels of these markers compared to that seen in the IL10- cells (even for the IL-10 to show the negative levels).

Reviewer #3: Minor Comments:

1. The introduction is overly detailed and could be condensed to 3-4 paragraphs, focusing on key background information relevant to the study’s primary findings.

2. The manuscript would benefit from representative flow cytometry dot plots accompanying the quantification graphs, improving clarity and interpretation of the data.

PLOS authors have the option to publish the peer review history of their article (what does this mean?). If published, this will include your full peer review and any attached files.

Reviewer #1: No

Reviewer #2: No

Reviewer #3: No

**Figure resubmission:****Reproducibility:** To enhance the reproducibility of your results, we recommend that authors of applicable studies deposit laboratory protocols in protocols.io, where a protocol can be assigned its own identifier (DOI) such that it can be cited independently in the future. Additionally, PLOS ONE offers an option to publish peer-reviewed clinical study protocols. Read more information on sharing protocols at https://plos.org/protocols?utm_medium=editorial-email&utm_source=authorletters&utm_campaign=protocols

---

## [Decision Letter · Decision Letter 1]

21 Aug 2025

PI3Kdelta-driven expansion of regulatory B cells impairs protective immune responses to Trypanosoma congolense parasite infection

PLOS Pathogens

Dear Dr. Marshall,

Thank you for submitting your manuscript to PLOS Pathogens. After careful consideration, we feel that it has merit but does not fully meet PLOS Pathogens's publication criteria as it currently stands. Therefore, we invite you to submit a revised version of the manuscript that addresses the points raised during the review process.

Please submit your revised manuscript within 60 days Oct 20 2025 11:59PM. If you will need more time than this to complete your revisions, please reply to this message or contact the journal office at plospathogens@plos.org. Please include the following items when submitting your revised manuscript:

We look forward to receiving your revised manuscript.

Kind regards,

Tracey J. Lamb

Section Editor

PLOS Pathogens

Tracey Lamb

Section Editor

PLOS Pathogens

Editor-in-Chief

PLOS Pathogens

orcid.org/0000-0003-2946-9497

Editor-in-Chief

PLOS Pathogens

orcid.org/0000-0002-7699-2064

**Additional Editor Comments:**

Comments from all reviewers for this revised manuscript are shown below for your information. Please add day 0 data to Figure 4 and a more detailed flow cytometry characterization of IL-10 producing populations in infected mice. Please also add representative plots for all data before resubmitting.

**Reviewers' Comments:**

Reviewer's Responses to Questions

**Part I - Summary**

Reviewer #2: In this revised manuscript, the authors examine the effects of B cell-specific expression of activated PI3Kd on responses to Trypanosome congolense. This is a solid descriptive report that demonstrates/confirms increased IL-10 producing cells and altered cytokines in mice expressing activated PI3Kd in B cells. It also further characterizes increased iNOS by macrophages in these B cell mice in response to T. congolense. While primarily correlative in its descriptions, it is a solid report. Although it is not clear that it provides much further insight into responses to T. congolense, it does provide insight into alterations associated with activated PI3Kd syndrome and is solid.

Reviewer #3: The revised version of the manuscript addressed all my comments and concerns, and it shows a great improvement from its original submission. I recommend acceptance on its current version.

Reviewer #4: This interesting study by Olayinka-Adefemi et al analyzes the role of regulatory B cells in Trypanosoma congolense infection generating by the gain of function mutation in PI3Kδ protein. The authors demonstrated that mice carrying this mutation has a higher number of IL10 secreting B cells and this impact in the control of infection and the develop of the immune response.

**Part II – Major Issues: Key Experiments Required for Acceptance**

Reviewer #2: Review PLOS Pathogens:

In this revised manuscript, the authors examine the effects of B cell-specific expression of activated PI3Kd on responses to Trypanosome congolense. This is a relatively descriptive paper which agrees with previous work (primarily Stark et al and Wray-Dutta et al) showing altered B cell populations with increased IL-10 “Breg” type cells in these mice and altered responses to infection. I was not one of the initial reviewers of this paper, but have evaluated the responses to the first set of reviews. The authors have tried to address the primary concern raised by multiple reviewers of the manuscript on how Bregs may contribute to the T. congolense response in these mice by performing IL-10 blocking experiments (Fig 9). While these experiments did not reverse parasite burden, they did alter some aspects of the immune response. The effect of IL-10 blockade was most pronounced on antibody class-switching, with some effects on iNOS production by macrophages. The authors’ interpretation of the limitations of this approach is reasonable, as there may be other mechanisms by which Bregs influence the anti-parasite response– the authors could strengthen their discussion by including other possibilities such as whether the altered Ig classes may contribute to the disease (and affect other immune parameters).

The other reviewer comments have also mostly been addressed or attempted by the authors. In addition to the day 0 data which is now included in Figure 5C, data at day 0 for Figure 4 would strengthen the manuscript. This would be especially beneficial for Figures 4A, D, and E, since there are already differences in IL-10 and IFN-g at the d4 timepoint. I agree that the authors should include representative flow plots for all data (for example, new data).

Additional comments include:

The authors suggest that altered cytokine with reduced IFNg:IL10 ratios contributes to the poor initial responses. However, in the IL-10 blocking experiments, they only look at day11, which is useful for the Ig class-switching and macrophage iNOS, but IFNg differs most at d4. Were there differences in cytokines earlier? Was TNF-a affected at d11 by blocking IL-10, as that was quite different at the later days in their original data?

If this was the initial review, additional comments include:

1. Given the hyper-IgM/reduced class switch antibody phenotype in the mutant mice, it is possible that qualitative differences in the early antibody response could contribute to the susceptibility to T. congolense. To address this possibility, the authors could cross the B cell-specific mutants to a strain incapable of producing soluble Ig or to a strain with an irrelevant BCR (the latter cross is relatively fast). This would remove soluble antibody specificity as a contributing factor to their observed phenotypes. However, as a re-review, this is not required.

2. The authors point out that the mutant mice have elevated IL-10 both at baseline and early during infection. Are differences in baseline IL-10 (or other factors) sufficient to impair T. congolense control, or does the Breg response to infection dictate susceptibility? Could this be addressed by depleting B cells at the time of infection? Alternatively, IL-10 blockade could be initiated at different timepoints shortly after infection, though this would not account for other effector molecules produced by Bregs. Again, I would have liked this if this was an initial review but would not require it for a re-review.

In summary, this is a solid descriptive report that demonstrates/confirms increased IL-10 producing cells and altered cytokines in mice expressing activated PI3Kd in B cells. It also further characterizes increased iNOS by macrophages in these B cell mice in response to T. congolense. While primarily correlative in its descriptions, it is a solid report. Although it is not clear that it provides much further insight into responses to T. congolense, it does provide insight into alterations associated with activated PI3Kd syndrome and is solid.

Reviewer #3: (No Response)

Reviewer #4: As a round 1 revised paper some important questions were answered but, however, I have some concerns.

1) The authors show that PI3Kδ hyperactivation in B cells is not associated with impaired germinal center or plasma cell responses, based on the cell surface markers analysis by flow cytometry. However, I believe that a histological analysis of germinal centers (at least of the spleen) is necessary, as extrafollicular structures have been described in similar parasites infections.

2) The characterization of different IL-10-producing cell populations was performed in uninfected mice. However, a more detailed characterization of these cells in infected mice using flow cytometry is still required.

**Part III – Minor Issues: Editorial and Data Presentation Modifications**

Reviewer #2: (No Response)

Reviewer #3: (No Response)

Reviewer #4: I realize that the introduction and discussion should be concise but the relevance of B cell and specially the Breg in other parasitic infection should be discussed.

How were identified de IL-10 secreting cell by flow cytometry in Figure S4? If they were identified by intracytoplasmic immunofluorescence, describe in Materials and Methods.

In the mRNA sequencing experiment, how many reads were analyzed per sample? Explain this point.

PLOS authors have the option to publish the peer review history of their article (what does this mean?). If published, this will include your full peer review and any attached files.

Reviewer #2: No

Reviewer #3: No

Reviewer #4: No

**Figure resubmission:**

**Reproducibility:**



---

## [Editor Report · Decision Letter 2]

23 Oct 2025

Dear Prf Marshall,

We are pleased to inform you that your manuscript 'PI3Kdelta-driven expansion of regulatory B cells impairs protective immune responses to Trypanosoma congolense parasite infection' has been provisionally accepted for publication in PLOS Pathogens.

Best regards,

Tracey J. Lamb

Section Editor

PLOS Pathogens

Tracey Lamb

Section Editor

PLOS Pathogens

Sumita Bhaduri-McIntosh

Editor-in-Chief

PLOS Pathogens

orcid.org/0000-0003-2946-9497

Michael Malim

Editor-in-Chief

PLOS Pathogens

orcid.org/0000-0002-7699-2064
---

## [Editor Report · Acceptance letter]

Dear Prf Marshall,

We are delighted to inform you that your manuscript, " 

PI3Kdelta-driven expansion of regulatory B cells impairs protective immune responses to Trypanosoma congolense parasite infection," has been formally accepted for publication in PLOS Pathogens.

Best regards,

Sumita Bhaduri-McIntosh

Editor-in-Chief

PLOS Pathogens

orcid.org/0000-0003-2946-9497

Michael Malim

Editor-in-Chief

PLOS Pathogens

orcid.org/0000-0002-7699-2064